# MixPatch: A New Method for Training Histopathology Image Classifiers

**DOI:** 10.3390/diagnostics12061493

**Published:** 2022-06-18

**Authors:** Youngjin Park, Mujin Kim, Murtaza Ashraf, Young Sin Ko, Mun Yong Yi

**Affiliations:** 1Department of Industrial & Systems Engineering, Korea Advanced Institute of Science and Technology, Daejeon 34141, Korea; youngjpark@kaist.ac.kr (Y.P.); mujinkm@kaist.ac.kr (M.K.); murtaza@kaist.ac.kr (M.A.); 2Pathology Center, Seegene Medical Foundation, Seoul 04805, Korea; noteasy@mf.seegene.com

**Keywords:** histopathology image analysis, deep learning, prediction uncertainty, confidence calibration

## Abstract

CNN-based image processing has been actively applied to histopathological analysis to detect and classify cancerous tumors automatically. However, CNN-based classifiers generally predict a label with overconfidence, which becomes a serious problem in the medical domain. The objective of this study is to propose a new training method, called MixPatch, designed to improve a CNN-based classifier by specifically addressing the prediction uncertainty problem and examine its effectiveness in improving diagnosis performance in the context of histopathological image analysis. MixPatch generates and uses a new sub-training dataset, which consists of mixed-patches and their predefined ground-truth labels, for every single mini-batch. Mixed-patches are generated using a small size of clean patches confirmed by pathologists while their ground-truth labels are defined using a proportion-based soft labeling method. Our results obtained using a large histopathological image dataset shows that the proposed method performs better and alleviates overconfidence more effectively than any other method examined in the study. More specifically, our model showed 97.06% accuracy, an increase of 1.6% to 12.18%, while achieving 0.76% of expected calibration error, a decrease of 0.6% to 6.3%, over the other models. By specifically considering the mixed-region variation characteristics of histopathology images, MixPatch augments the extant mixed image methods for medical image analysis in which prediction uncertainty is a crucial issue. The proposed method provides a new way to systematically alleviate the overconfidence problem of CNN-based classifiers and improve their prediction accuracy, contributing toward more calibrated and reliable histopathology image analysis.

## 1. Introduction

For the past decade, deep learning (DL) has been widely applied in computer vision tasks and achieved impressive performance, primarily due to the rapid development of convolutional neural network (CNN) techniques. The automatic diagnosis of heterogeneous diseases that can lead to loss of life is a challenging application for DL techniques. Cancer is a highly heterogeneous disease and one of the leading causes of death, ranking second in deaths per year in the world [1]. To diagnose the presence of cancer, pathologists usually examine whole-slide images (WSIs) to identify abnormal cells. The growth in the number of yearly cancer cases has led to expert pathologists working long hours, thereby increasing the chance of human errors, which has been found to be approximately 3% to 9% in anatomical pathology [2]. To alleviate this problem, DL-based frameworks for WSI analysis have been developed to assist pathologists [3,4,5,6].

DL-based WSI analysis involves the handling of large WSIs [6,7], each of which consists of many gigapixels (typically 50,000 × 50,000 pixels). Given such a large size, it is difficult to input a WSI into a CNN model due to computational constraints. Additionally, reducing the resolution of a WSI for CNN model training can negatively affect model performance because the WSI information is distorted [8]. To overcome this challenge, researchers have proposed patch-based frameworks for WSI analysis using DL [9,10,11,12]. Such frameworks commonly consist of three phases for WSI analysis: (1) splitting the target WSI into patches, (2) extracting features from these patches using a patch-level classifier, and (3) identifying abnormalities in the WSI by aggregating the extracted features of patches [13]. Prior research on patch-based analysis focused on how to design an overall framework. In particular, previous studies concentrated on how to aggregate the extracted features of patches to identify abnormalities in WSIs. However, in addition to the proper design of an overall framework, the effective training of a patch-level classifier is of critical importance because the performance of the patch-level classifier is the foundation of an overall framework.

To extract the features of patches, patch-level classifiers have been trained based on transfer learning, with little attention given to the characteristics of patches [3,4,5,14,15,16,17]. Additionally, to improve the performance of a CNN model as a patch-level classifier, prior studies employed image modification techniques such as data augmentation [18,19], color transformation [20,21], and stain normalization [22,23,24]. The goals of image modification techniques are to amplify the number of patch images, extract the morphological features, and reduce the deviations across WSI scan devices. Despite these diverse efforts, prediction uncertainty has not received much attention in patch-based analysis even though it is a serious issue, particularly in the medical domain. In this study, we propose a novel method, called MixPatch, that actively considers prediction uncertainty associated with histopathology patches.

Prediction uncertainty is largely indicated by the confidence level of the prediction output from a CNN model. A critical issue in the current baseline approach is that the confidence level is given on a binary scale of 0 or 1, thus creating overconfidence problems [25,26]. More specifically, most abnormal histopathology patches are mixed with benign regions and nonbenign regions [27]. Extracted patches are labeled by pathologists to build a training dataset for patch-level classifiers. In this process, if an extracted patch includes various class regions, the extracted patch is labeled according to the most serious diagnosis by a pathologist. However, most of the abnormal patches are mixed with benign regions and nonbenign regions to varying degrees. This *mixed-region variation* property is difficult for patch-level classifiers properly to consider. For example, if a small area of a patch is nonbenign, the prediction uncertainty of the case should be high, as most of the cell is benign. However, because of the overconfident nature of CNN, a patch-level classifier trained with a traditional method will produce a confidence value of 1 or very close to 1, even for this highly uncertain case. To alleviate this overconfidence problem, a patch classifier needs to be trained by properly incorporating the mixed-region variations in histopathology images. If prediction uncertainty information for mixed regions could be properly applied in the training process, the parameters of the CNN model would be more effectively trained, effectively enriching the extracted features of patch-based information and ultimately contributing to enhanced overall performance of the framework.

The objective of this study is to propose a new training method, called MixPatch, to improve patch-level classifiers by specifically addressing the prediction uncertainty problem and to examine its effectiveness in improving diagnosis performance in the context of histopathological image analysis. The central objective of the proposed MixPatch method is to build a new subtraining dataset that has a predefined mix of benign vs. nonbenign patches in certain ratios and the associated ground-truth labels. MixPatch is designed to explicitly consider the mixed-region variations in histopathological patch images. The dataset is generated using a small size of confirmed, clean (benign and nonbenign) histopathological patches. To define a new ground-truth label, proportion-based soft labeling [28] is used. MixPatch is a novel method applicable to the training of CNN models in the domain of digital pathology. As described in Figure 1, MixPatch prevents or limits the overconfidence problem by explicitly addressing the high level of prediction uncertainty associated with highly mixed-region cases in histopathological images.

The major contributions of this paper are as follows:We propose a new method designed to train a CNN-based histopathology patch-level classifier. The method is applicable to many medical domains in which patch-based images are used.The proposed method estimates prediction uncertainty to varying degrees to enrich the extracted features of patch-based information and improve the overall performance of the framework for WSI analysis.The proposed method is tested based on histopathology stomach datasets to assess the performance improvements achieved in comparison with other state-of-the-art methods at the patch level and slide level.

## 2. Literature Review

### 2.1. Patch-Based WSI Analysis

Participation in grand challenges for digital pathology (https://grand-challenge.org/, accessed on 13 June 2022) has led to remarkable developments in automatic diagnosis. In particular, WSI classification has received extensive attention from research communities. Most researchers have relied on patch-based classification approaches due to the computational limitations of directly applying CNN models for WSI analysis. In each competition, patch-based approaches have been among the best performers.

The existing patch-based digital pathology frameworks consist of patch-level classifiers and WSI-level classifiers. A patch-level classifier is responsible for classifying each patch based on a respective class label. In contrast, the WSI-level classifier considers various information, such as the features extracted from patches, the locations of patches, and the number of patches in aggregation, to obtain a final decision with regard to the slide in question. Thus, given the complexity of this approach, the current frameworks are primarily concerned with the design of the WSI-level classifier. For example, a study focused on developing a framework that enabled CNNs to efficiently analyze WSIs by incorporating multiple instance learning was proposed [29]. Additionally, a top-performing team in the grand challenge proposed a binary classification framework in which 11 types of features were first extracted based on the available morphological and geometrical information, and then these features were used for classification with a random forest classifier [30]. Although their study relied on traditional machine learning approaches for classification modeling, recent studies have predominantly proposed frameworks using DL. Wang et al. [13] proposed a DL-based WSI multiclassification framework that first selects discriminative patches, extracts features for each class using a patch-level classifier, and then utilizes the extracted features to diagnose diseases using a multi-instance deep learning network. Dov et al. [31] proposed weakly supervised instance learning for whole-slide cytopathology images with unique slide structures. Duran-Lopez et al. [32] proposed a novel aggregated CNN model for slide-level classification using the patch-level classes obtained from a CNN. Li et al. [33] proposed a multiresolution multi-instance learning model to detect suspicious regions for fine-scale grade prediction.

The design of an overall framework is an important issue, and the tiling process (i.e., creating patches from a WSI) and patch-level classification are the fundamental building blocks of these frameworks. To implement the tiling process, the extant frameworks employed image modification methods [6,30]. The goals of such methods are to increase the amount of data using rotation, to extract morphological features using different color scales, and to reduce the variation in dyeing or scanning. Additionally, most existing studies trained patch-level classifiers by applying transfer learning, metric learning, and fine-tuning methods based on existing CNN architectures such as ResNet, VGG, and DenseNet [33,34,35,36,37,38]. These studies focused on improving the performance of patch-level classifiers in different ways, but did not pay attention to the issue of prediction uncertainty. It is important to address prediction uncertainty because a patch-level classifier is utilized as a feature extractor. Properly incorporating prediction uncertainty into the training process can substantially enrich the extracted features of patch-based information, thereby positively influencing the performance of the applied WSI analysis framework.

### 2.2. Uncertainty in Deep Learning

CNN models have displayed state-of-the-art performances in many image classification tasks [39,40,41,42]. Although CNN-based approaches have achieved superior performance in various applications over the past decade, CNN models tend to predict labels with overconfidence [43,44]. For example, CNN models often produce a high confidence probability of 91%, even for ambiguous cases and public datasets [45]. Incorrect predictions with overconfidence can be harmful. It is essential for the probability of the predicted label to reflect the corresponding likelihood of ground-truth correctness. This consideration is especially important when a CNN model is applied to a medical dataset [26].

As a remedy to this problem, two approaches have been proposed: uncertainty quantification and confidence calibration. The first approach estimates uncertainty based on a probability density over all outcomes. Bayesian probabilistic deep learning [43] and MC (Monte Carlo) dropout with ensembles [44] are two common uncertainty quantification approaches. However, such methods have not been widely adopted due to implementation challenges and long training times [46]. The second approach measures prediction uncertainty with values of confidence. The confidence level is the highest value from a probability distribution that can be extracted from the softmax layer. Methods based on the second approach can provide appropriately calibrated confidence information to limit the overconfidence issue. The second approach, the confidence-based uncertainty measurement approach (also called the confidence calibration approach), is more suitable for medical applications than is the first approach. In general, the classification of labels for medical applications are associated with the N-stage in pathology. Although the first approach separately produces a predicted label and the corresponding uncertainty, the second approach tries to produce a confidence probability for each stage and selects the predicted label with the highest confidence probability. The confidence probability for each label is helpful for computer-aided diagnosis. Additionally, the second approach is more straightforward than the first approach, and some methods that rely on the second approach, such as excessive dropout, do not use intentional random noise. Thus, robust CNN models can be established.

Noise distributions are commonly used in confidence calibration [28,47,48]. However, applying intentional random noise can cause problems for histopathological patch classification. Taking a different approach without intentional random noise, several methods utilize an additional subtraining dataset to increase variability in the training process [49]. The basic objective of this approach is to build a new subtraining dataset that consists of mixed images and their new ground-truth labels. Specifically, a new mixed image is a combination of two or more images, and the corresponding ground-truth label is defined using a label smoothing method based on the mix combination. For example, if images A and B are mixed at the same ratio, the ground-true label is based on a weight of 0.5 for both categories of A and B. Multiple methods have been proposed to mix images, including MixUp [50], CutMix [51], and RICAP [52]. MixUp combines two images by overlaying them and redefining a new ground-truth label to create a new subtraining dataset. CutMix replaces part of an image with a cropped patch from another training image and redefines a new ground truth label based on the proportions of the respective image areas. RICAP combines four images randomly cropped according to boundary positions and redefines a new ground-truth label with the same image area proportions.

The performance of these image mixing methods has been evaluated using public image datasets such as MNIST [53], CIFAR10 [54], and ImageNet [55]. In public image datasets, the main target is placed over the center of the image so that most of the main target exists during the cropping process [56]. However, these methods have the potential to cause problems when applied to histopathological images. Specifically, the cropping process can easily produce mislabeled data if nonbenign areas are all cropped from an uncertain abnormal patch. This paper proposes a novel method that produces improved performance in handling prediction uncertainty by considering the mixed-region variation in histopathological patches. The new method builds and uses an additional subtraining dataset as a patch-level classifier. The dataset consists of mixed patches, each of which is a set of mixed small images, and no cropping is required; additionally, the corresponding ground-truth labels are determined based on the mixing ratio.

## 3. Method

The primary goal of the proposed method is to address the problem of prediction uncertainty by utilizing a prearranged set of mixed patches. This method generates a new subtraining dataset consisting of randomly drawn mixed patches and their ground-truth labels and applies them to the model training process, which is further illustrated in Figure 2.

### 3.1. A New Subtraining Dataset: Mixed Patches and Their Ground-Truth Labels

The essential component of the proposed method is a new subtraining dataset. The dataset consists of mixed patches and their ground-truth labels. The generation process for the mixed patches and their ground truth labels is as follows. Let $\left(x,y\right)\in D_{original}$, $\left(x_{mixed},y_{mixed}\right)\in D_{mixed-patch}$, and $\left(x_{mini},y_{mini}\right)\in D_{mini}$ denote the original dataset, a new subtraining dataset, and a minipatch dataset, respectively. To build a new subtraining image $x_{mixed}$, minipatches $x_{mini}$ are concatenated. We use $D_{mini}$ to eliminate the cropping process and build a new mixed patch because the cropping process is not appropriate for histopathological images; this approach reduces the probability that noise affects the dataset. We initialize the number of minipatch images k to build a single $x_{mixed}$. The sizes of $x_{mini}$ and $x_{mixed}$ can be adjusted according to the parameter k. The number of cases for a single $x_{mixed}$ is _|Dmin|_$P_{k}=\left|D_{mini}\right|!/\left(\left|D_{mini}\right|-k\right)!$, indicating that an enormous number of $x_{mixed}$ values can be generated. Thus, a data augmentation effect is achieved.

After generating a new subtraining image $x_{mixed}$, we define a new ground-truth label $y_{mixed}$. As demonstrated by several existing methods [50,51,52], new ground-truth labels play an important role in achieving high performance and producing high calibration confidence. In prior work, new ground-truth labels were defined based on the proportions of the regions of the images. For example, prior studies defined a new ground label with a weight of 0.5 for each class if a mixed image included cats and dogs in the same proportion. However, histopathological images differ from the images found in public datasets. Histopathological images have to be diagnosed as abnormal if any proportion of the mixed image contains abnormalities. Additionally, even if a mixed image is diagnosed as abnormal, the confidence should not be fixed at 1, because the underlying composition of the classes in the image is diverse, reflecting the mixed-region variation property. Thus, to overcome the overconfidence problem, for any abnormal mixed patch, the value of abnormality in a new ground-true label needs to be defined from 0.5 to 1 according to the proportions of normal and abnormal minipatches in a mixed patch.

### 3.2. Training Process

The subtraining dataset generated from the above process is used to train a patch-level classifier. Many existing methods for confidence calibration generate new subtraining datasets, divide the data into multiple minibatches, and periodically insert selected minibatches into the training process (e.g., [51]). However, given the context of medical image analysis, our method takes a more cautious, conservative approach of mixing the newly generated subtraining dataset with the original dataset (as opposed to using only the newly generated subtraining dataset) for every minibatch. Specifically, our approach builds a set of minibatches, each of which is based on a combination of the randomly sampled original dataset and the newly generated subtraining dataset in a certain prefixed proportion according to the parameter γ(0≤γ≤1). Additionally, the combined minibatches are used throughout the whole training process. Furthermore, we define loss functions as follows:(1)$${ℒ}_{\;original}=\sum_{i=1}^{\left|ℬ\right|\times \left(1-\gamma \right)}D_{KL}(f\left(x_{i}\right)||y_{i})$$
(2)$${ℒ}_{Mixed-Patch}=\sum_{i=1}^{\left|ℬ\right|\times \gamma }D_{KL}(f\left(x_{i}^{Mixed-Patch}\right)||y_{i}^{Mixed-Patch})$$
(3)$${ℒ}_{Total}=w{ℒ}_{\;original}+\left(1-w\right){ℒ}_{\;Mixed-Patch}$$
where |ℬ| is the size of the minibatch; f is a classifier; $D_{KL}$ is the Kullback–Leibler divergence function; $\left(x_{i},\;y_{i}\right)\in D_{original}$ is the original training dataset; and w (0≤w≤1) is the weight for the loss of the raw training data.

### 3.3. Data Rebalancing

A new ground-truth label for a mixed patch is defined as abnormal even if a single abnormal minipatch is included. When four minipatches are used to form a single mixed patch, the probability of the new ground-truth label being defined as normal is one in sixteen (2^4^) because all four minipatches must be normal, meaning that most of the mixed patches are likely to be designated as abnormal, resulting in a data imbalance problem. Techniques for solving data imbalance problems have been presented in various studies [57]. In this study, we employ a data resampling technique to solve the data imbalance problem. This method involves creating a balanced minibatch based on the probability of extracting an individual class from an existing dataset.

## 4. Experiment

### 4.1. Dataset

We constructed a new large histopathology dataset extracted from stomach WSIs obtained at Seegene Medical Foundation, which is one of the largest diagnosis and pathology institutions in South Korea. These slides were stained with hematoxylin and eosin and scanned by a Panoramic Flash250 III scanner at 200× magnification. The data were collected by the Seegene Medical Foundation, and their use for research was approved by the Institutional Review Board (SMF-IRB-2020-007) of the organization as well as by the Institutional Review Board (KAIST-IRB-20-379) of the Korea Advanced Institute of Science and Technology (KAIST), the university that collaborated with the medical foundation. Informed consent to use their tissue samples for clinical purposes was obtained from the medical foundation’s designated collection centers. All experiments were performed in accordance with the relevant guidelines and regulations provided by the two review boards. All patient records were completely anonymized, and all the images were kept and analyzed only on the company server.

For an original training dataset, we collected 486 WSIs from different patients, and the images consisted of 204 normal and 282 abnormal slides that were classified and independently confirmed by two pathologists (Table 1). The extracted patch dataset consisted of 32,063 normal and 38,492 abnormal patches. For a minipatch dataset, we used the same WSIs used for the original training dataset, but the tiling size was one-quarter. The minipatch dataset consisted of 3500 randomly selected normal and 3500 abnormal minipatches. For a test dataset, we collected 98 WSIs from different patients, and the images included 48 normal and 50 abnormal slides. The test dataset consisted of 3733 normal and 3780 abnormal patches.

### 4.2. Implementation Details

The proposed method was implemented in Python with the PyTorch library on a server equipped with 2 NVIDIA RTX 2080 TI GPUs. We used ResNet-18 as the backbone CNN architecture. The primary goal of this study was to analyze the impact of the proposed methodology, not to produce the highest performance. Thus, we thought it would be better to compare the effects of the proposed methodology by adopting a contemporary, light CNN architecture. The CNN classifier was trained with the Adam optimizer [58] and $\beta_{1},\;\beta_{2}\;$, and the decay coefficient were set to 0.9, 0.999, and 0.001. We trained models with 2 GPUs and set the minibatch size to 128. The models were trained for 60 epochs and used an initial learning rate of 0.1, which was divided by 10 at 20 and 40 epochs.

### 4.3. Comparison of Methods

To assess the effectiveness of the proposed method, we compared five models, each of which was trained using a different method (Table 2): Baseline, Label Smoothing (LS), Cutout, CutMix, and MixPatch (proposed method). Table 2 provides a summary of key differences of these methods, each of which is further detailed below.

***Baseline:*** The baseline method uses transfer learning and fine-tuning, which are commonly utilized by patch-level classifiers. The baseline method trains a model using hard labeling with a one-hot-encoded label vector, for which the ground-truth label value is specified as 1 and other labels are 0; thus, the model is designed to predict a label with 100% certainty [59]. For this reason, a model trained with the baseline method has the possibility of experiencing overconfidence issues. No data augmentation is employed in this method.

***Label******smoothing (LS)*** is a simple regularization method designed to alleviate the overconfidence problem. The LS method assigns the highest value of confidence (lower than 1) to the ground-truth class and low values from noise distributions (higher than 0) to all of the classes with a parameter α, as shown below:$$y_{k}^{ls}=y_{k}\left(1-\;\alpha \right)+\;\alpha /K$$
where *k* is the *k*th class, *K* is the total number of classes, and α is the smoothing parameter.

For evaluation, α was set to 0.2 in this study. As in the baseline method, no data augmentation is employed in this method.

***Cutout*** is a region dropout-based regularization method. Cutout randomly masks square regions of an image during training. This training method exhibited excellent robustness and performance [60]. However, Cutout may remove informative regions from training images. Thus, this method may generate mislabeled data. Cutout must define the size of pixels that are removed from an input image. This study defined the pixel size as a quarter of the image size based on the setting used in a previous study [60].

***CutMix*** has been used as a state-of-the-art method for region dropout. CutMix performs data augmentation for improved accuracy and implements soft labeling for confidence calibration. CutMix builds a new training image by attaching a cropped portion of another image to a region of image that is removed and uses the soft labeling technique in consideration of the mix proportion of the new training image. Based on the labeling rules in histopathology, CutMix may generate mislabeled data. For example, as shown in Table 2, an image with small abnormal regions is attached to a base normal image, and it will be predicted as normal when the true label is abnormal.

***MixPatch*** is the proposed method. MixPatch achieves a data augmentation effect similar to that of other region dropout methods, and ratio-based soft labeling is employed for confidence calibration. However, MixPatch will not accidently produce mislabeled training data, which is a strength when compared with other region dropout methods. MixPatch incorporates a soft labeling technique for confidence calibration and considers unique image combinations and labeling rules, which are specifically established for histopathological images. In our experiment, the value of abnormality for a new ground-truth label is defined as a constant that increases from 0.6 to 0.9 according to the abnormal patch ratio in a mixed patch (Table 3). Weighted random sampling, a data resampling technique, is employed for data rebalancing. We set the parameter γ to 0.3. There is no difference between the weights of the original data and the weights of the new subtraining data used to calculate the loss value, meaning that the parameter w was set to 0.5.

### 4.4. Evaluation Metrics

For evaluation, this study uses accuracy, sensitivity, specificity, area under a receiver operating characteristic curve (AUROC), and expected calibration error (ECE). Accuracy is the main metric for the performance of image classifiers, but it is not informative enough for medical systems. AUROC is a metric for binary classification in consideration of sensitivity and specificity. This study defined confidence value as the variable for AUROC analysis, as in prior research [61]. AUROC is a vital evaluation criterion for understanding the performance of models for automatic diagnosis systems as it shows how good the diagnostic model is at distinguishing between positive and negative classes by considering net benefit (sensitivity) over diagnostic cost (1-specificity). ECE has been used as the primary empirical metric to measure confidence calibration. ECE is a metric of how much confidence in predictions reflects actual model accuracy and a small value of ECE indicates a small difference between output confidence and model accuracy—small degree of miscalibration.

True positive (*TP*) is the correct classification of the positive class (Table 4). For example, the model classifies the patch as abnormal if a patch contains cancerous cells. True negative (*TN*) is the correct classification of the negative class. For example, when there is no cancerous cell present in the patch, the model predicts the patch as normal. False positive (*FP*) is the incorrect prediction of the positives. For example, the patch does have cancerous cells, but the model classifies the patch as abnormal. False negative (*FN*) is the incorrect prediction of the negatives. For example, there are cancerous cells present in the patch, and the model predicts the patch as normal.


**Accuracy**


It is the rate of correct identification of all items:$$\mathrm{Accuracy}\;=\frac{TP+TN}{TP+TN+FP+FN}$$


**Specificity**


It is the rate of correct identification of negative items:$$\mathrm{Specificity}\;=\frac{TN}{TN+FP}$$


**Sensitivity**


It is the rate of correct identification of positive items:$$\mathrm{Sensitivity}\;=\frac{TP}{TP+FN}$$


**Receiver Operating Characteristic Curve (ROC-Curve)**


The receiver operating characteristic curve (ROC-curve) represents the performance of the proposed model based on a threshold. In this study, we defined the confidence score of positive defined as the threshold. It is the graph of True Positive Rate (TPR) vs. False Positive Rate (FPR).
$$\mathrm{TPR}\;=\frac{TP}{TP+FN}$$
$$\mathrm{FPR}\;=\frac{FP}{FP+TN}$$


**Area Under the ROC Curve (AUROC)**


AUROC provides the area under the ROC-curve integrated from (0, 0) to (1, 1). It measures performance based on all classification thresholds. AUROC has a range from 0 to 1.


**Expected Calibration Error (ECE)**


ECE is approximated through partitioning predictions into equally spaced bins B and taking a weighted average of the bins’ accuracy vs. confidence difference. More precisely,
$$\mathrm{ECE}=\overset{M}{\underset{m=1}{\sum }}\frac{\left|B_{m}\right|}{n}\left|accuracy\left(B_{m}\right)-confidence\left(B_{m}\right)\right|$$
where n is the number of samples, and M is the number of bins, $B_{m}$ is the set of samples whose prediction confidence falls into the interval $I_{m}=\left(\frac{m-1}{M},\frac{m}{M}\right]$.

## 5. Results

The performances of the training methods were assessed by analyzing the mean and standard deviation of accuracy, sensitivity, specificity, AUROC, and ECE obtained from the five models trained in each method. The performance results for the trained models are shown in Table 5, ROC curve is shown in Figure 3, and detailed information on the ECE is shown in Figure 4.

As shown in Table 5, the proposed method, MixPatch, yields the best performance in accuracy, sensitivity, specificity, AUROC, and ECE among the five models examined. The LS method does not show any advantage compared to the baseline method. The LS method attempts to fit training cases with a 0.9 confidence level, thus producing many test cases distributed in the bin of 0.85–0.95 (Table 6); the results suggest that the model is 90% sure about the results of most cases, even for cases that are very clear. This phenomenon is not suitable from the perspective of confidence calibration, so it is understandable that ECE performance deteriorates. The Cutout method uses one-hot encoding, similar to the baseline method. Cutout exhibits a higher ECE than the baseline approach because the Cutout method does not use a confidence calibration method, although the accuracy of this approach is comparatively low. The CutMix method yields a slightly higher ECE result than the baseline method, probably because of the influence of ratio-based soft labeling; however, the accuracy and AUROC decrease slightly because of the possibility of mislabeling. The proposed method, MixPatch, shows increased classification performances and decreased ECE, which are both desirable. Thus, applying soft labeling combined with the mix ratio of the images according to the MixPath labeling rules makes a positive contribution to both classification performance and confidence calibration.

Furthermore, we illustrate the specific ECE results of the compared methods with a reliability diagram. In Figure 4, ground truth represents the ideal scores for the confidence calibration methods. The confidence value of a prediction should reflect its accuracy. Among the compared methods, CutMix and MixPatch yield similar values that are closest to the ground truth, indicating that ratio-based soft labeling methods are effective for confidence calibration.

In addition to the quantitative analysis using the ECE metric, we examine confidence distributions by quantifying true and false predictions for test cases to determine how well the proposed method considers prediction uncertainty (Table 6). A skew to a high confidence value is desired for the confidence distribution in the cases of true predictions. In contrast, a skew to a low confidence value is desired for the confidence distribution in the cases of false predictions. We need to carefully examine confidence distributions for cases with false predictions to understand the effects of the proposed methods in terms of prediction uncertainty.

The models trained with the baseline and Cutout methods exhibit an overconfidence issue (see red bins in Table 6). The two models produce high confidence values, even for false predictions. Thus, these methods should not be used when the confidence value is used as a threshold for decision making and are not suitable as patch-level classifiers, particularly in the context of histopathological image analysis. The model trained using LS or CutMix yields a flatter distribution than the baseline model for false predictions, indicating that this method better alleviates overconfidence and produces lower confidence values for uncertain cases. The model trained using MixPatch produces a flat distribution that is similar to the distribution obtained with LS or CutMix, indicating that the proposed method can effectively deal with overconfidence issues. Additionally, the proposed method, MixPatch, exhibits better performance than the other methods, confirming that the method is more suitable than the other methods for building histopathology patch-level classifiers.

For further analysis of the effect of applying confidence calibration, we construct confusion matrixes according to the relevant threshold values (Table 7). We define the confidence value for abnormalities as an indicator. The baseline classification threshold is 0.5 because binary classification is used. Typical methods for WSI classification are based on counting the labels of patch-level predictions. For this method, a threshold for a patch-level classifier plays an important role in WSI classification. For example, if a low threshold is applied, a WSI classification framework will be very sensitive to positive results.

For all of the compared methods, the lower the threshold is, the lower the false-negative ratio, and the higher the false-positive ratio, with some notable differences in accuracy. For example, in the MixPatch model, if 0.1 is defined as the threshold value, the WSI classification framework is very sensitive to positive (i.e., abnormal) values while maintaining high accuracy. Conversely, in the LS model, if the threshold is defined as 0.1, it is sensitive to positive values, but the model predicts most of the results as abnormal, resulting in low accuracy.

For qualitative analysis, we applied Grad-CAM to uncertain patch images. In the first case (see Figure 5), it seems that all models can find the abnormal locations and predict them correctly. Overall, the activation map of other methods other than the baseline method is dispersed widely. However, in the case of MixPatch, the size of the activation map does not increase, which we believe is due to the confidence calibration effect. As MixPatch uses an image that combined normal and abnormal patches, it seems that MixPatch method wants to train a model more clearly to distinguish between normal regions and abnormal regions. Therefore, the activation map appears to be smaller than other methods.

The second case is more difficult than the first case. All models except MixPatch have activations on both of the normal and abnormal regions. Especially difficult regions in the second case are the second and third quadrants. The second quadrants contain the dark and cellular areas, mimicking poorly differentiated carcinoma; however, it is lymphoid aggregates. The third quadrant shows a very small part of suspicious glandular epithelium, and slightly distorted normal parietal cells. All models predict this patch as abnormal. However, in the activation map, such difficult regions made the comparison models all confused about separating abnormal regions from normal regions. On the other hand, the MixPatch model shows noticeable improvement in clearly distinguishing abnormal regions from normal regions.

The objective of the patch-level classifier is to extract important information from patches for WSI classification. MixPatch not only increases the performance of patch-level prediction, but also produces appropriate prediction uncertainty values through confidence calibration. Therefore, for WSI classification, we applied an existing method [63] that uses confidence values rather than a simple method of counting patch-level predictions. This method uses a CNN model and a feature cube. A feature cube is generated using the predicted confidence scores of each label from patches. A CNN model is used as a slide-level classifier, and feature cubes are used as inputs for the slide-level classifier. In this study, we trained five CNN models under the same conditions as considered for the patch-level classifier, and Resnet-18 was used in each approach. Slides used to train patch-level classifiers were also used to train slide-level classifiers. Additionally, to analyze the performance of the slide-level classifiers using an independent set of slides at a large scale, we used separately collected, annotated test slides, including 459 normal and 604 abnormal slides.

As presented in Table 8, MixPatch produced a 1.5% performance improvement compared to the baseline at the slide level. The difference of 1.5% is notable when this approach is practically applied in the medical domain. LS yields a higher ECE than the baseline, but its WSI classification performance is similar to that of the baseline. The reason why LS yields a high ECE value is that many cases are assigned a high confidence value close to 0.9, which is the maximum confidence level for the LS slide-level classifier. Further, as shown in Table 6, LS generates more alleviated confidence scores for uncertain cases (false predictions). Thus, despite the increased ECE, it seems that the WSI classification performance of LS did not deteriorate much compared to that of the baseline, due to the more effective control of overconfidence. For CutMix, the accuracy of the patch-level classifier is lower than that of the baseline, but the slide-level classification performance is higher, probably due to better handling of overconfidence. Consistent with the study results obtained at the patch level, MixPatch exhibits the best performance at the slide level among the five classification methods considered.

## 6. Discussion

The objective of this study was to explore the possibility of improving the performance of a patch-level classifier by developing a new DL training approach called MixPatch, which employs a set of mixed patches in predefined mixing ratios and their associated labels, within the context of histopathological image analysis. The study results confirm the superiority of the proposed approach when compared to the existing approaches, not only at the patch level but also at the slide level. Prior studies have proposed two-step frameworks, each of which consists of a patch-level classifier and a slide-level classifier. The performance of a patch-level classifier is the foundation of those frameworks. However, such frameworks utilize transfer learning and well-known CNN architectures for patch-level classifiers without considering the specific characteristics of patches or the corresponding prediction uncertainty. In this study, we propose a new method for training a patch-level classifier specifically designed to address the mixed-region variation inherent in histopathological images and the derived patches.

A significant factor that underlies the performance of MixPatch is the effect of performing data augmentation without mislabeled data. A small number of minipatches can be used to build a vast number of single mixed patches, resulting in numerous different mixed patches. In general, deep learning models perform better as the amount of available data increases. Furthermore, the proposed method can solve the overconfidence issue related to prediction uncertainty when a patch-level classifier is trained. Addressing the prediction uncertainty of patch-level classification should be an important part of WSI classification frameworks. The WSI-level classifier determines whether to trust each patch’s prediction based on its estimation of prediction uncertainty. Therefore, a patch-level classifier that appropriately handles prediction uncertainty should be used in a WSI classification framework to help it make more calibrated decisions.

The method proposed in this study has some limitations and boundary conditions that need to be noted. To build a single mixed patch, we utilized 128 × 128 pixel minipatches; this size is the minimum required for pathologists to make diagnosis decisions at the patch level. Additionally, we utilized four minipatches to build a single mixed patch. In future studies, a sensitivity analysis could be conducted using various subtraining datasets that consist of mixed patches with 9 or 16 minipatches or different pixel sizes. To define new ground-truth labels, we considered a constant increase in labels from 0.6 to 0.9 based on the proportion of abnormal minipatches in a mixed patch. However, labels could be defined differently by employing a different labeling scheme, such as an exponential scheme. In this study, we defined the proportion of the new subtraining dataset in the minibatch to be 0.25. In future studies, this percentage could be adjusted, and a sensitivity analysis could be performed to find the optimal value.

## 7. Conclusions

In this study, we have proposed a new method, MixPatch, designed to train a CNN-based histopathological patch-level classifier. The proposed method is the first that considers confidence calibration for prediction uncertainty when training a patch-level classifier. Given that the performance of the patch-level classifier is the foundation of overall framework performance, the proposed method should be used to improve the performance of existing frameworks. Moreover, it should be noted that the proposed method improves the performance of the patch-level classifier by addressing prediction uncertainty, which is particularly important in the domain of medical image analysis, where prediction uncertainty is a crucial issue. The proposed approach provides a new way to systematically alleviate overconfidence problems without a performance degradation, compared with the extant methods. The confidence calibration method proposed in this study is an important step toward securing a completely reliable diagnose performance of histopathological image analysis.

## Figures and Tables

**Figure 1 diagnostics-12-01493-f001:**
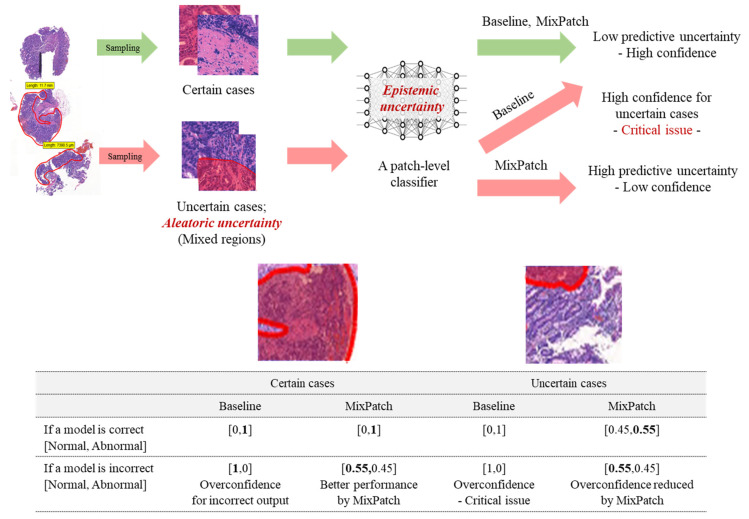
Baseline vs. MixPatch. A single WSI generates multiple patches. The process of tiling creates certain case patches and uncertain case patches. Most parts of a certain patch are covered by a single label, but those of an uncertain patch are mixed. The baseline methods are overconfident, even for uncertain patches and incorrect outputs. The proposed method, MixPatch, overcomes these problems by explicitly incorporating the mixed-region variations in histopathological images into the training process.

**Figure 2 diagnostics-12-01493-f002:**
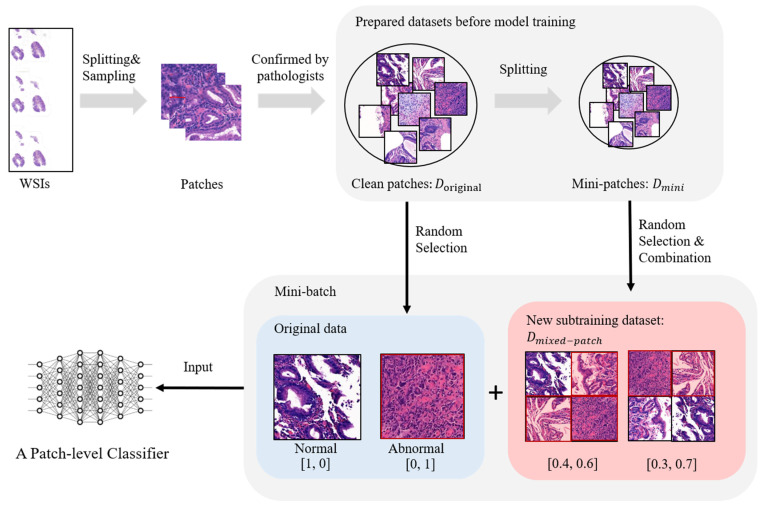
The overall process of the proposed method. In the existing methods, the patch-level classifier is trained using a CNN model and a cleaned patch dataset, $D_{original}$, which pathologists previously confirmed. The proposed method, MixPatch, additionally uses a new subtraining dataset, which consists of image $x_{mixed}$ and label $y_{mixed}$. $x_{mixed}$ is built by combining randomly selected images from the minipatch dataset. $y_{mixed}$ is defined according to the ratio of abnormal mini-patches. In the figure, a minibatch is a randomly built mix of samples from $D_{original}$ and samples from $D_{mixed-patch}$.

**Figure 3 diagnostics-12-01493-f003:**
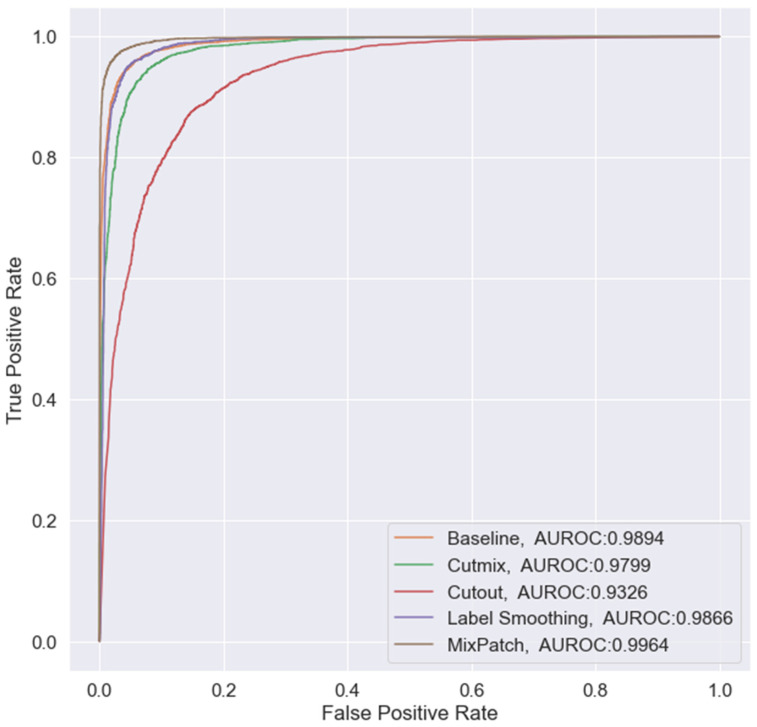
ROC curve for the different methods.

**Figure 4 diagnostics-12-01493-f004:**
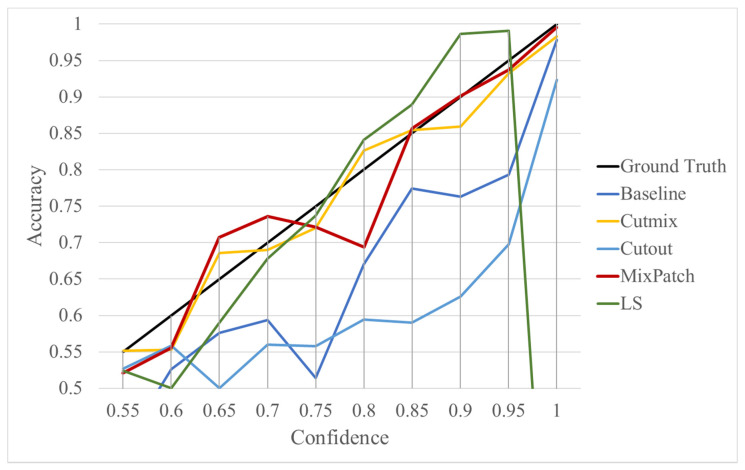
Integrated reliability diagram for patch-level classifiers trained using each method.

**Figure 5 diagnostics-12-01493-f005:**
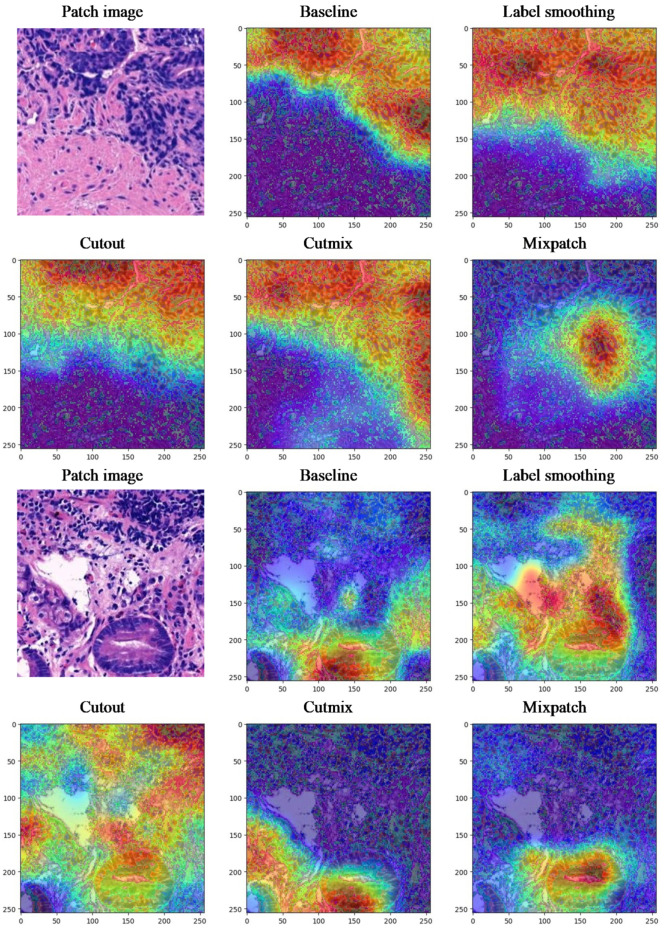
The Grad-Cam [62] visualization examples for uncertain patch images.

**Table 1 diagnostics-12-01493-t001:** Compositions of datasets.

	Original Training Dataset (256 × 256)		Minipatch Dataset (128 × 128)		Test Dataset (256 × 256)	
Class	Normal	Abnormal	Normal	Abnormal	Normal	Abnormal
WSIs	204	282	204	282	48	50
Patches	32,063	38,492	3500	3500	3733	3780

**Table 2 diagnostics-12-01493-t002:** Summary of the compared methods.

	Baseline	LS	Cutout	CutMix	MixPatch
Data augmentation	X	X	O	O	O
Soft labeling	X	O	X	O	O
Ratio reflection	X	X	X	O	O
All correct labeling	O	O	X	X	O
Image	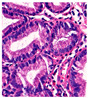	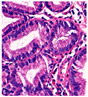	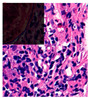	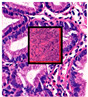	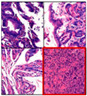
Label	Normal 1.0	Normal 0.9Abnormal 0.1	Abnormal 1.0	Normal 0.8Abnormal 0.2	Normal 0.4Abnormal 0.6
Actual label	Normal	Normal	Abnormal	Abnormal	Abnormal

**Table 3 diagnostics-12-01493-t003:** Labeling strategy for a mixed patch.

Abnormal Patch Ratio in a Mixed Patch	New Ground-Truth Label for a Mixed Patch
0/4	[0.9, 0.1]
1/4	[0.4, 0.6]
2/4	[0.3, 0.7]
3/4	[0.2, 0.8]
4/4	[0.1, 0.9]

**Table 4 diagnostics-12-01493-t004:** The confusion matrix for outcome of predictions.

		Actual	
		Abnormal (Positive)	Normal (Negative)
Prediction	Abnormal (Positive)	True positive (*TP*)	False positive (*FP*)
	Normal (Negative)	False negative (*FN*)	True negative (*TN*)

**Table 5 diagnostics-12-01493-t005:** Performance comparison of the alternative methods.

Training Methods	Accuracy ↑ (In Percent)	Sensitivity ↑ (In Percent)	Specificity ↑ (In Percent)	AUROC ↑	ECE ↓ (In Percent)
Baseline	95.46 ± 0.79	96.96 ± 1.15	93.95 ± 0.71	0.9914 ± 0.0027	1.83 ± 0.43
LS	94.76 ± 0.94	96.15 ± 1.43	93.35 ± 0.51	0.9861 ± 0.0038	6.62 ± 0.34
Cutout	84.88 ± 0.47	82.33 ± 0.86	87.46 ± 0.31	0.9289 ± 0.0027	7.06 ± 0.28
CutMix	93.70 ± 0.91	94.30 ± 1.19	93.11 ± 0.92	0.9826 ± 0.0041	1.36 ± 0.22
MixPatch	97.06 ± 0.27	97.65 ± 0.23	96.46 ± 0.48	0.9958 ± 0.0006	0.76 ± 0.18

**Table 6 diagnostics-12-01493-t006:** Confidence distributions of each method.

Methods	Confidence Distributions	
	False Predictions	True Predictions
Baseline	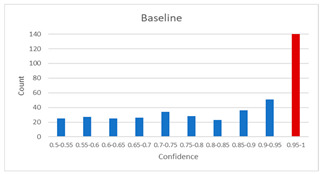	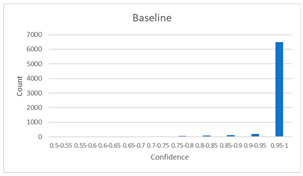
Label smoothing	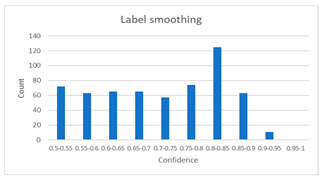	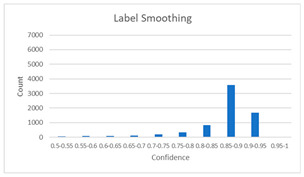
CutMix	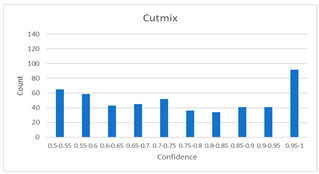	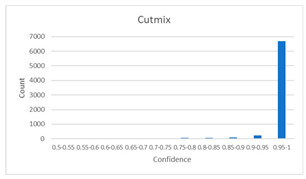
Cutout	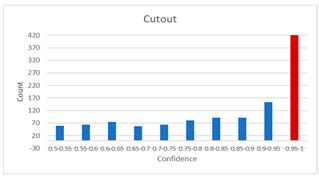	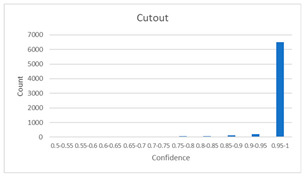
MixPatch	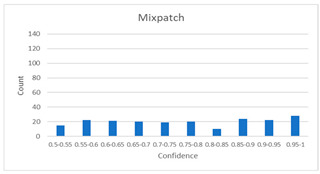	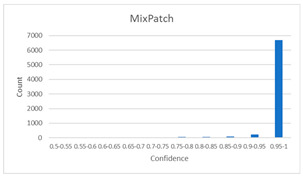

**Table 7 diagnostics-12-01493-t007:** Confusion matrix for each method with a threshold approach (X = prediction, Y = true).

	$\mathrm{Threshold}\;(If\;Cofindence{}_{AB}\geq Threshold,\;\;Then\;Prediction=Abnormal)$				
Model	0.5 (Baseline)	0.4	0.3	0.2	0.1
Baseline	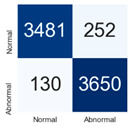	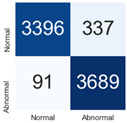	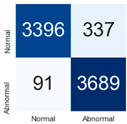	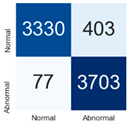	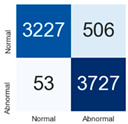
LS	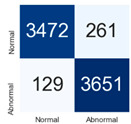	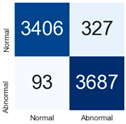	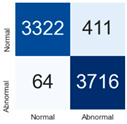	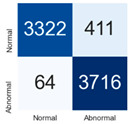	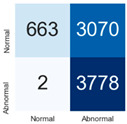
CutMix	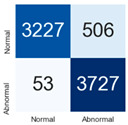	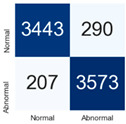	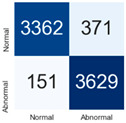	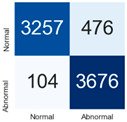	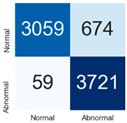
Cutout	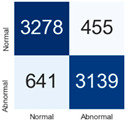	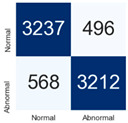	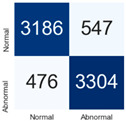	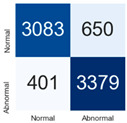	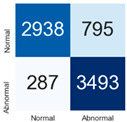
MixPatch	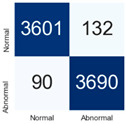	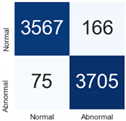	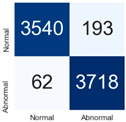	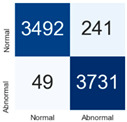	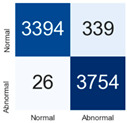

**Table 8 diagnostics-12-01493-t008:** Performance in WSI classification.

WSI Classifiers	WSI-Level Accuracy ↑ (In Percent)
Baseline	97.06 ± 0.29
LS	97.15 ± 0.18
Cutout	95.82 ± 0.57
CutMix	97.46 ± 0.18
MixPatch	98.53 ± 0.16

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
