# Peer review of "MixPatch: A New Method for Training Histopathology Image Classifiers"

_diagnostics, 2022, doi:10.3390/diagnostics12061493_

Round 1

Reviewer 1 Report

The authors presented a method for training patch-level CNN to classify histopathology images. The idea is interesting and worthy. However, the superiority of the proposed model needs further validation. I have the following suggestions for authors (Comments are not in the order of importance).

Comments

  1. Why is Resnet-18 chosen as the backbone CNN? Clarification is needed.
  2. Some comparative data need to be shown in the abstract. Also, the abstract needs a thorough rewrite.
  3. Patch level diagnosis needs to be visually extracted.
  4. The “Conclusion” section is missing in the manuscript.
  5. The sentence “CNN models have displayed state-of-the-art performances in many image classification tasks” can be written as “CNN models have displayed state-of-the-art performances in many image classification tasks [1][2]. ([1] DOI: 10.1371/journal.pone.0264586 [2]: DOI: 1038/s41598-021-03287-8)
  6. The related work can be summarized in the form of a table.
  7. The proposed method needs to be validated on public benchmark datasets such as BACH to corroborate its superiority.

Author Response

Responses to Reviewer 1 Comments

Comments and Suggestions for Authors
The authors presented a method for training patch-level CNN to 
classify histopathology images. The idea is interesting and 
worthy. However, the superiority of the proposed model needs 
further validation. I have the following suggestions for 
authors (Comments are not in the order of importance).
We thank you for taking the time to review the manuscript and provide constructive and 
insightful comments. We have revised the paper to fully address the comments raised within 
the revision time available, and certainly believe that these revisions are positive enhancements 
to this paper. We have also thoroughly revised the manuscript for improved clarity and 
readability. Detailed responses are provided below.
Note: The reviewer comments are presented in courier font, and our responses are in 
roman font.
Comments
1. Why is Resnet-18 chosen as the backbone CNN? 
Clarification is needed.
We also thought a lot about which CNN architecture to use as the backbone model. A 
variety of complex and high-performance CNN architectures have been proposed;
however, the primary goal of this study was to analyze the impact of the proposed 
methodology, not to produce the highest performance. Thus, we thought it would be 
better to compare the effects of the proposed methodology by adopting a 
contemporary, light CNN architecture. As such, we decided to use Resnet-18 as the
backbone CNN architecture, which showed sufficiently high accuracy in the baseline 
model with its simple architecture.
We have added the sentence “We used ResNet-18 as the backbone CNN architecture. The 
primary goal of this study was to analyze the impact of the proposed methodology, not to 
produce the highest performance. Thus, we thought it would be better to compare the effects 
of the proposed methodology by adopting a contemporary, light CNN architecture.” in the 
implementation details section (p. 11).
2. Some comparative data need to be shown in the abstract. 
Also, the abstract needs a thorough rewrite.
Thank you for the suggestion. We have modified the abstract to include comparative 
data as follows:
“More specifically, our model showed 97.06% accuracy, an increase of 1.6% to 
12.18%, while achieving 0.76% of expected calibration error, a decrease of 0.6% to 
6.3%, over the other models.”
Also, we have thoroughly revised the abstract.
3. Patch level diagnosis needs to be visually extracted.
We applied Grad-CAM to the patch-level visual analysis and conducted a qualitative 
analysis based on activation maps extracted using Grad-CAM (see Figure 5 on page 
22).
4. The “Conclusion” section is missing in the manuscript.
Thank you. As suggested, we have added the "Conclusion” section.
5. The sentence “CNN models have displayed state-of-the-art 
performances in many image classification tasks” can be 
written as “CNN models have displayed state-of-the-art 
performances in many image classification tasks [1][2]. 
([1] DOI: 10.1371/journal.pone.0264586 [2]: DOI: 
1038/s41598-021-03287-8)
We have changed the sentence to add the references (p. 6). Thank you.
6. The related work can be summarized in the form of a 
table.
The related work directly relevant to the proposed method is summarized in the form 
of a table in Table 2 (p. 12), located within “4.3. Comparison of methods.” Those
methods applied image conversion or soft labeling to confidence calibration.
7. The proposed method needs to be validated on public 
benchmark datasets such as BACH to corroborate its 
superiority.
The proposed method in this study uses a clean mini-patch dataset to create one 
mixed-patch. A comparison study might be conducted using a public benchmark 
dataset. However, creating a mini-patch dataset by decomposing an image-level 
labeled benchmark dataset is very likely to generate a noise-labeled mini-patch as the 
benchmark datasets are based on image-level labels, as shown below:
Public benchmark datasets 
BACH 
• Image-level annotated microscopy dataset (train 400, test 100)
• Pixel-wise annotated whole-slide image (train 30, test 10)
BreakHis
• Image-level annotated image (benign:2480, Malignant:5429)
PatchCamelyon
• Image-level annotated image 327.680 color images (96 x 96px)
One exception is BACH. It provides a pixel-wise annotated whole-slide image
dataset. However, the dataset is too small to ensure proper training and testing of the 
patch-level classifier in this study, which used 486 WSIs. Further, in order to create a 
clean mini-patch using BACH, a group of pathologists must newly annotate the
dataset and generate a mini-patch dataset. Also, to annotate different tissue of organs, 
we need to have pathologists in that field. These issues are all very challenging within
the given revision time frame. Furthermore, creating annotations using the 
pathologists we recruit will stain the characteristics of public benchmark data.
Therefore, we did not use BACH in this revision.

Reviewer 2 Report

This paper presented a MixPatch method, which is a data-augmentation-based CNN method for patch-level histopathological classifying. This paper is organized well, while major revision is still needed. 

1. Can you compare this designed method with the conventional CNN method without data augmentation to demonstrate the advantages of this designed method?

2. How did you choose the weight for the designed method, the selection method of the weight and the comparison of different weights should be discussed.

3. The comparison study is also needed for different training and testing datasets setup. 

4. Figures in table 5 and table 6 should have axis titles. Also, the figures need to be more clear. 

5. Figure 4 is not clear.

Author Response

Responses to Reviewer 2 Comments

Comments and Suggestions for Authors
This paper presented a MixPatch method, which is a dataaugmentation-based CNN method for patch-level 
histopathological classifying. This paper is organized well, 
while major revision is still needed.
We thank you for taking the time to review the manuscript and provide constructive and 
insightful comments. We have revised the paper to fully address the comments raised within 
the revision time available, and certainly believe that these revisions are positive enhancements 
to this paper. We have also thoroughly revised the manuscript for improved clarity and 
readability. Detailed responses are provided below.
Note: The reviewer comments are presented in courier font, and our responses are in 
roman font.
Comments
1. Can you compare this designed method with the 
conventional CNN method without data augmentation to 
demonstrate the advantages of this designed method?
In this study, data augmentation techniques (cropping, flipping, Rotation, etc.) are not 
applied to any model. In particular, the baseline method and LS do not use any data 
augmentation method. CutMix, Cutout, and Mixpatch use a new sub-training dataset, 
thereby producing a data augmentation effect. Details are provided in “4.3. 
Comparison of methods” (pp. 12-13). Thus, by comparing the baseline model with the 
MixPatch model (e.g., Table 5), we show the advantages of this designed method over 
the conventional method without data augmentation. Table 2 includes this information
(p. 12).
2. How did you choose the weight for the designed method, 
the selection method of the weight and the comparison of 
different weights should be discussed.
In this study, we defined the proportion of the new subtraining dataset in the 
minibatch to be 0.25. Sensitivity analysis, which finds the optimal value of the 
parameter, is not considered in the current study. Given the scope of the study, we 
have mentioned it in the “Discussion” section as future work (p. 24).
3. The comparison study is also needed for different 
training and testing datasets setup.
The proposed method in this study uses a clean mini-patch dataset to create one 
mixed-patch. A comparison study might be conducted using a public benchmark 
dataset. However, creating a mini-patch dataset by decomposing an image-level 
labeled benchmark dataset is very likely to generate a noise-labeled mini-patch as the 
benchmark datasets are based on image-level labels, as shown below:
Public benchmark datasets 
BACH 
• Image-level annotated microscopy dataset (train 400, test 100)
• Pixel-wise annotated whole-slide image (train 30, test 10)
BreakHis
• Image-level annotated image (benign:2480, Malignant:5429)
PatchCamelyon
• Image-level annotated image 327.680 color images (96 x 96px)
One exception is BACH. It provides a pixel-wise annotated whole-slide image 
dataset. However, the dataset is too small to ensure proper training and testing of the 
patch-level classifier in this study, which used 486 WSIs. Further, in order to create a 
clean mini-patch using BACH, a group of pathologists must newly annotate the 
dataset and generate a mini-patch dataset. Also, to annotate different tissue of organs, 
we need to have pathologists in that field. These issues are all very challenging within
the given revision time frame. Furthermore, creating annotations using the 
pathologists we recruit will stain the characteristics of public benchmark data. 
Therefore, we did not use BACH in this revision. 
4. Figures in table 5 and table 6 should have axis titles. 
Also, the figures need to be more clear.
Thank you for the constructive feedback. The figures in Table 5 (now, Table 6) and in 
Table 6 (now, Table 7) have now axis titles. Also, for improved clarity, we have 
regenerated the figures presented in Table 6 (p. 19) and reformatted the figures 
presented in Table 7 (p. 20). 
5. Figure 4 is not clear.
In fact, in our previous manuscript, Figure 3 and Figure 4 were redundant as Figure 4 
showed the same results in a different format, separated for each individual model. 
Thus, to prevent any confusion and improve the clarity of the message, we have 
deleted Figure 4. Thank you

Round 2

Reviewer 1 Report

The authors attempted to address the issue raised in the previous round. However, some of the comments from the previous revision seems not addressed and the changes in the revised manuscript are also not highlighted which makes it very hard to match the revision and review comments point by point.  For example, a comparison with the state-of-the-art methods is still not so persuasive. Please take it seriously and correct all of them and highlighted the change with track-change.
